

# Development And Application of WRF(v4.1.2)-uEMEP(v5) Model at the City with the Highest Industrial Density: A Case Study of Foshan

Liting Yang[1], Ming Chang[1,*], Shuping Situ[2], Weiwen Wang[1], Xuemei Wang[1]

[1]Guangdong-Hongkong-Macau Joint Laboratory of Collaborative Innovation for Environmental
Quality, Institute for Environmental and Climate Research, Jinan University, Guangzhou, 511443, China
[2]Foshan Ecological and Environmental Monitoring Station of Guangdong Province, Foshan, 528000, China

*Correspondence to*: Ming Chang (changming@email.jnu.edu.cn)

**Abstract.**
The study aims to develop and apply the WRF-uEMEP model to simulate air quality at the city scale, with a focus on Foshan, the city with the highest industrial density. The interaction between atmospheric diffusion and chemical reactions in different regions further complicates the modeling process. Therefore, this study proposes a multi-scale approach to build an urban air quality model with a
resolution of 250 meters by integrating different models. The model takes into account the effects of urban structure and takes into account atmospheric diffusion and chemical reactions in different regions. The research process included model development, calibration, and validation using existing air quality data in Foshan. The study shows that the WRF-uEMEP model effectively captures the impact of urban structure on air pollutant processes. The simulation results reveal the spatial and temporal distribution of
air pollutants in Foshan, providing valuable insights for urban air quality management.

## 1 Introduction

Due to the limitations of traditional ground-based observations on spatial and temporal scales, the study of regional air quality is mostly carried out with the help of numerical models. At present, there is very
little model that can meet all the requirements of urban air quality management, and the air quality model at the regional/city scale can reflect the degree of influence of urban structure (building volume and road network, etc.) on physical and chemical processes such as the transport, transformation and deposition of air pollutants, but it also needs to take into account the interaction of atmospheric diffusion and chemical reactions between regions/countries and regions/countries, so the establishment
of urban air quality models at the scale of 100 meters requires the combination of different scale models(Anjali and Rao, 2011; Chen et al., 2011).
At present, multi-scale air quality models have been applied in many urban areas to meet the needs of urban air quality management. The Enviro-HIRLAM-M2UE model was tested for multi-scale air pollution and emergency protection in the Copenhagen region of Denmark, and the results of the single-



layer nested multi-scale model were satisfactory, but the resolution of the microscale model was recommended to be 3-5 times higher than that of the previous model(Baklanov and Nuterman, 2009). CMAQ-ADMS(Roads) uses WRF to input the meteorological field to CMAQ, and the CMAQ simulation grid is reduced to 20 meters by bilinear interpolation, and then combined with the ADMS (Roads) model to simulate the nitrogen oxide concentration on the roads in London, England, but there

is a certain error in the location of the model simulation close to the road(Beevers et al., 2012). Street-in-Grid (SinG) combines a street network model, an urban network model of cross-canyons and roads (MUNICH) and a chemical transport model Polair3D to simulate the concentration of nitrogen oxides in the suburbs of Paris, but the new model fails to demonstrate the ability to assess background concentrations compared to traditional chemical transport models(Kim et al., 2018). The Modair4health

system, which includes the online model WRF-Chem and the fluid mechanics model VADIS, uses linear and nonlinear methods to calculate morbidity and mortality in order to quantify the health effects of air pollution, and is an important multi-scale model tool for comprehensive air quality and health assessment in the city of Coimbra, Portugal (Silveira et al., 2023). KC-TRAQS combines the diffusion model and local air quality measurement datasets to estimate the relative contribution of air pollution in

Kansas City, USA(Isakov et al., 2019), and similarly, following the processing and analysis of feasible data procedures, the LUR method is applied in many countries and cities, such as Hong Kong, Taipei-Keelung Metropolitan Area, Barcelona, Spain, and San Francisco, USA, but can only reflect the air pollution during the period of use (Bai et al., 2023; Criado et al., 2022; Li et al., 2021; Li et al., 2023). The GEM-MACH-PAH was used in Toronto, Canada, to explore the contribution of motor vehicles to

benzene and polycyclic aromatic hydrocarbons in ambient air, but the uncertainty of atmospheric chemistry is difficult to assess(Whaley et al., 2020). HYCAMR combines the community model CAMx and the block-scale diffusion model R-LINE to estimate pollutant concentrations in the near-road environment, but does not take into account the emission factors of different roads(Parvez and Wagstrom, 2019). The CAIRDIO-Les large vortex model simulates micrometeorology and air pollution

in the Dresden Basin, Germany(Weger and Heinold, 2023). The VEIN-MUNICH model was developed and applied to the road-dense city of São Paulo, Brazil, to predict NOx emissions from road vehicles(Gavidia-Calderón et al., 2021). Multi-scale air quality models have been effectively developed and applied in urban air quality management, and can increasingly meet the requirements of urban air quality management.

The uEMEP model is based on the well-known Gaussian principle, and the uniqueness of the model lies in the fact that it performs the "local fraction" calculation included in the EMEP model, which allows the uEMEP model to be embedded in the EMEP model, making the model suitable for high-resolution calculations, and the uEMEP model has been successfully applied in Norway, Poland and other countries and regions(Maciej Kryza, 2022; Denby et al., 2020). Located in the Pearl River Delta region

of Guangdong Province, Foshan is an important component city of the Guangdong-Hong Kong-Macao Greater Bay Area, a typical manufacturing city in the developed coastal areas of southern China, and was once known as the manufacturing capital of the world(Liu et al., 2023). Figure 1 reveal that, compared with the above-mentioned countries/regions that have applied the 100-meter-scale urban air quality model, Foshan is in a period of rapid urbanization and industrialization, due to its high

population density and complex road network, and the complex urban building structure and large number of motor vehicle exhaust emissions have made the air quality in Foshan an unprecedented



challenge. Therefore, this study intends to use the mesoscale meteorological model WRF combined with the air quality downscaling model uEMEP to achieve high-resolution air quality modeling at the block scale in Foshan City, and to better understand the performance of the 100-meter scale numerical
model. It helps people to grasp the air pollution situation in urban areas more accurately, so as to put forward more scientific and targeted action plans to alleviate air pollution.

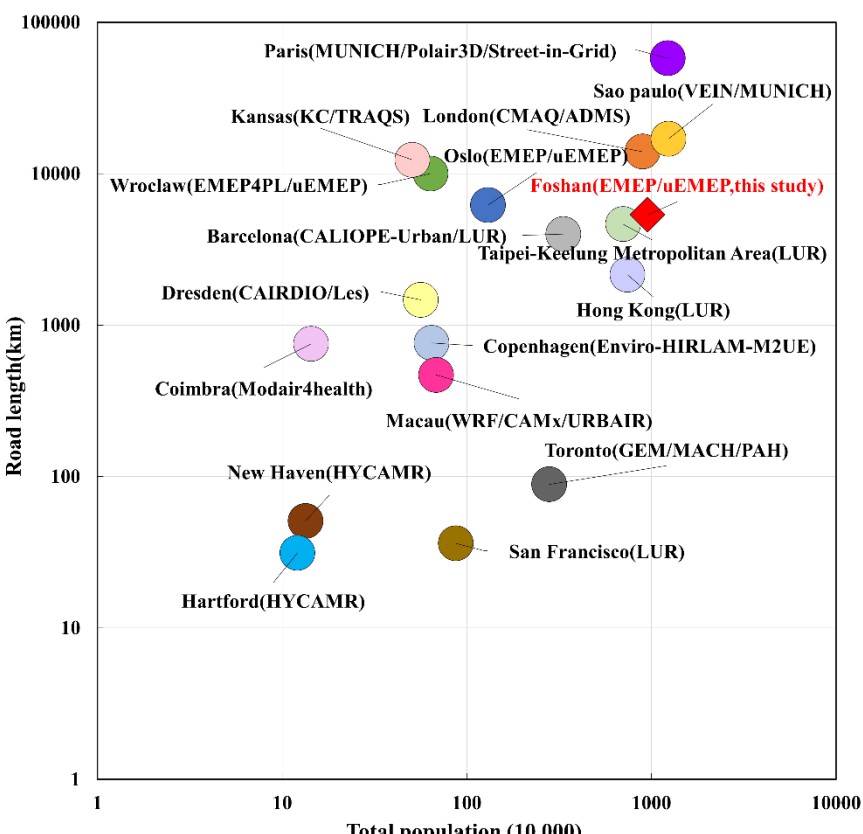

**Figure 1: Urban population and road network conditions using street resolution prediction models.**

## 2 Methodology

### 2.1 The WRF-EMEP-uEMEP model description

In this study, the regional weather prediction model WRF (v4.1.2), the chemical transport model EMEP (rv4_33) and the downscaling chemical transport model uEMEP (v5) were used to simulate the spatial
and temporal distributions of nitrogen oxides, particulate matter and ozone concentrations in Foshan. As shown in Fig. 2(a), the WRF model covers the entire Pearl River Delta with two layers of nesting, the number of simulated grids is set to 130×110, 136×151, the horizontal grid resolution is 9km×9km, 3km×3km, the model top pressure is set to 100 hPa, and the vertical direction is divided into 30 layers.



As shown in Fig. 2(b), the study area, mesh number, and mesh resolution of the EMEP model are the
same as those of the second layer of the WRF model. As shown in Fig. 2(c), the mesh range of the
uEMEP model covers the city of Foshan, and the number of simulated meshes is set to 480×480 and the
horizontal resolution is 250m×250m.
The WRF model weather field is driven by global 1°×1° reanalysis data
(http://rda.ucar.edu/datasets/ds083.2/, FNL) and the land use information uses data GLC2020 the
European Space Agency (ESA). Table 1 shows the physical parameterization and chemical protocols of
WRF-EMEP-uEMEP.

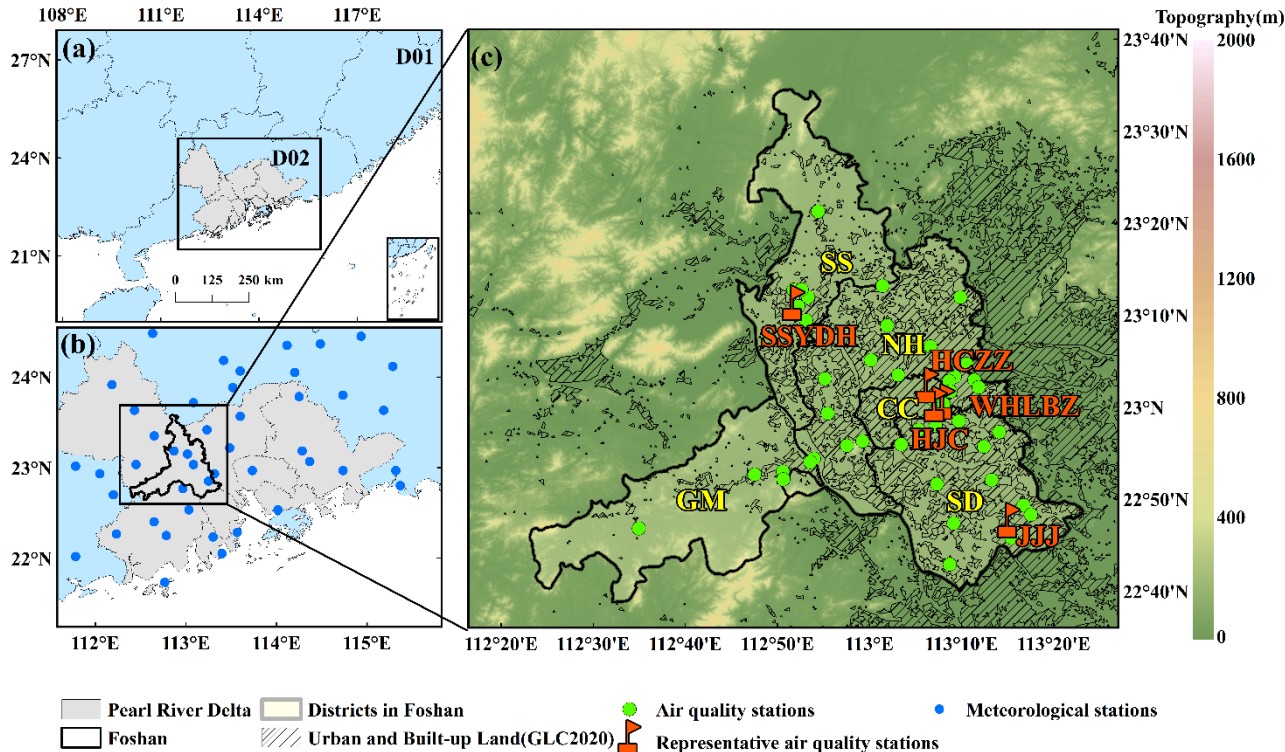

Figure 2: Model domains and locations of 33 air quality monitoring stations (green dots), 47 meteorological stations (blue dots),
and 5 representative air quality monitoring stations (red flags). The figure on the right shows the elevation of the terrain (m). The
5 representative air quality monitoring stations are Sanshuiyundonghai (SSYDH, suburban station), Jinjiju (JJJ, industrial
station), Huacaizhizhong (HCZZ, urban-developed station), Huijingcheng (HJC, urban-developing station), and
Wenhualubianzhan (WHLBZ, roadside station).The 5 districts in Foshan are Sanshui district(SS), Nanhai district(NH),
Chancheng district(CC), Gaoming district(GM), and Shunde district(SD).(a)WRF mode domain settings;(b)EMEP mode domain
settings;(c)uEMEP mode domain settings.

Table 1: WRF-EMEP-uEMEP simulation physical and chemical parameterization scheme settings.

| Scheme | Processes | Parameterization |
|---|---|---|
| **Physical** | Microphysics | Morrison double-moment |





| | Longwave Radiation | RRTMG scheme |
| --- | --- | --- |
| | Shortwave Radiation | RRTMG shortwave |
| | Surface Layer | Eta similarity |
| | Land Surface | Noah-MP Land Surface Model |
| | Urban Surface | BEM |
| | Planetary Boundary layer | Mellor-Yamada-Janjic scheme |
| | Cumulus Parameterization | Grell-3 |
| **Chemical** | Chemism | EmChem09 |

## 2.2 Introduction to uEMEP downscaling methods

The uEMEP mode can be run using two downscaling methods, both of which utilize a Gaussian diffusion model to simulate the concentration of contaminants at high resolution. The choice of downscaling method will depend on high-resolution emissions data, the first of which is the emissions redistribution method, which means that only the following types of high-resolution emissions data are available, such as population density, road network data, or land-use data. The second downscaling
method, the independent emission method, is available in both uEMEP and EMEP modes of input high spatiotemporal resolution emission inventories, and the gridded emissions data are fully consistent with the local emissions data, i.e., the proxy data is given in the form of emissions and summarized into the CTM grid emissions, and the two methods are equivalent(Mu et al., 2022). Based on the emission redistribution method, the EMEP-uEMEP model can be used to simulate the air quality at the scale of
100-meter urban blocks, and effectively simulate the diffusion of ozone precursors and particulate matter at the scale of urban blocks, which has broad application prospects.

## 2.3 The Development methods for localized emissions inventories

In order to accurately simulate air quality at multiple scales, it is necessary not only to ensure that the boundary layer conditions and background meteorological environment and other relevant variables in
the local scale air quality model can be accurately identified by the microscale model, but also to respond to multiple scale emission inventories and other required multi-scale inventories (Russo et al., 2019). The production of localized multi-scale emission inventories often requires a lot of manpower and material resources, and it is difficult to carry out large-scale investigation and production. Due to the difficulty of localizing the preparation and preparation of the list, it is difficult to deploy and apply
the block-scale model in China.
All input files required for the EMEP model (except aircraft emissions data and forest fire emission data) can be downloaded from the EMEP open source website, including land use data, road dust data, emission source data, time allocation factor data, settlement data, etc. (Simpson et al., 2012). However, the original land use data and emission data provided by the EMEP model are only applicable to the
European region, so the land use data and emission source data need to be updated locally. In this study, the 2017 China Monthly Average Multi-Resolution Emissions Inventory (http://meicmodel.org.cn,MEIC) data was used to replace the European emission source data in EMEP.



The SNAP classification method is used to input the emission inventory in the EMEP model, and the pollution source types in the MEIC 2017 emission data are divided into five categories: fixed
combustion sources, residential sources, industrial sources, transportation sources, and agricultural sources. The formula for converting a MEIC list to a SNAP list is as follows Eq. (1):

$$E_o^i = \sum_1^j (E_o^j \times f_o^j) \tag{1}$$

where o=1,2,...O; O is the pollutant type; i=1,2,...I; I is the number of SNAP classifications; j=1,2,...J; J is the MEIC classification number; $E_o^j$ is the amount of each pollutant in the MEIC inventory; $f_o^j$ is the
allocation coefficient of each pollutant in the MEIC inventory; and $E_o^i$ is the SNAP classification benchmark of each pollutant.

The formula for calculating pollutants in the EMEP model is as follows Eq. (2):

$$E_t^i = E_o^i \times f_{m.PRD}^i \times f_{h.PRD}^i \tag{2}$$

where i=1, 2, ...I; I is the number of SNAP classifications; t is a certain time; $f_{m.PRD}^i$ is the local
monthly time allocation factor in the Pearl River Delta; $f_{h.PRD}^i$ is the local hourly time allocation factor in the Pearl River Delta; $E_t^i$ is the calculated amount of pollutants.

The main types of emission data that need to be prepared for uEMEP are: road mobile sources, industrial point sources, shipping emission sources and residential sources. For the preparation of Foshan's local emission data, this study uses OpenStreetMap (OSM)(Openstreetmap Contributors,
2020) road network data, uses localized road weights to obtain traffic exhaust emission data, and uses the Global Human Settlement Layer (http://data.europa.eu/89h/2ff68a52-5b5b-4a22-8f40-c41da8332cfe) 250m grid population data replaces the residential wood burning emission data, the localized shipping emission data of the Pearl River Delta is used, and the shipping emission data is NOx emissions, covering the main rivers and shipping ports of the Pearl River Delta, and the industrial point
source emission data of Foshan is used to reduce the emission data of NOx and particulate matter(Fig. 3).




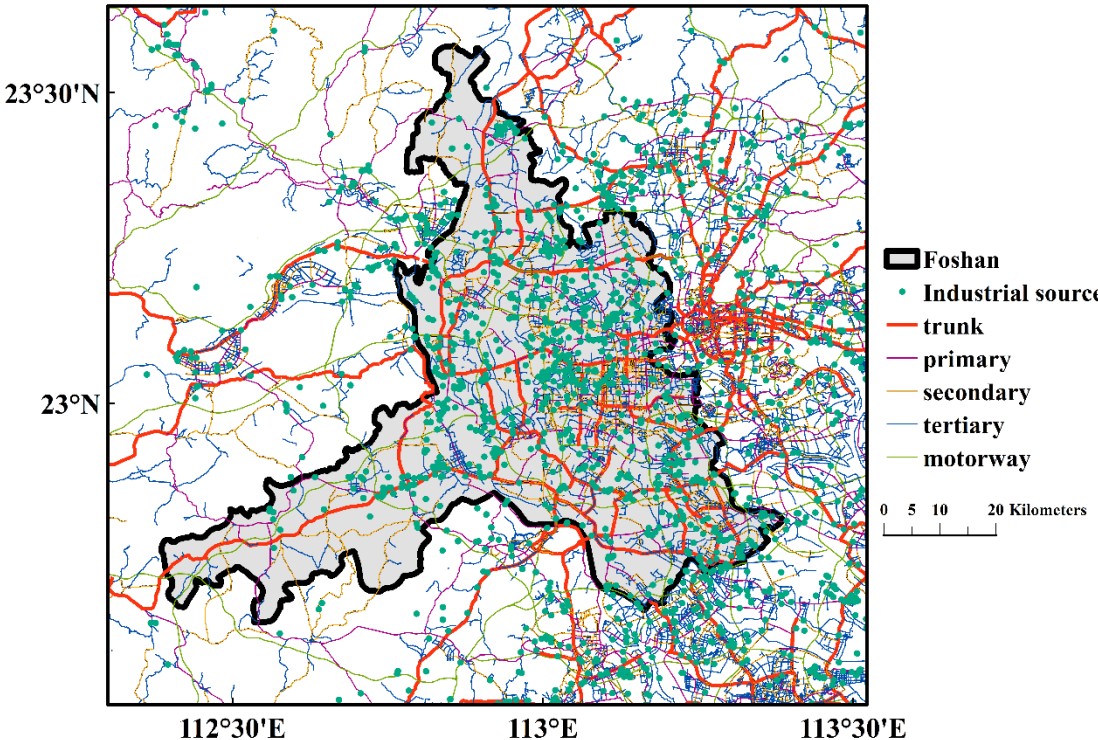

**Figure 3: Spatial distribution of road network and industrial point sources in Foshan, China. Road network map :**
**©OpenStreetMap, distributed under the Open Data Commons Open Database License (ODbL) v1.0.**

## 2.4 Measurement Datasets

The meteorological observation data used in this study are the hourly observation data of China's surface weather stations in the Pearl River Delta in 2021 (https://data.cma.cn/), with a total of 47
meteorological stations, and the mesoscale and block-scale pollutant data are based on the conventional pollutant observation data of the Guangdong Air Quality Forecasting and Early Warning Center (http://113.108.142.147:20032/Home/Index), with a total of 38 ambient air quality monitoring stations. The distribution of meteorological and ambient air quality monitoring stations is shown in Fig. 2.

## 2.5 Case classification

In this study, the only four $NO_2$ pollution cases in Foshan in 2021 were subjectively classified by weather patterns and simulation experiments were carried out (Yan et al., 2021). The L1 period is January 13, 2021 - January 16, 2021 Beijing time, during which $NO_2$ and $PM_{2.5}$ are combined pollution, and the weather situation is high-pressure control and frontal influence. The L2 period is January 19, 2021 ~ January 20, 2021 Beijing time, and the weather period is $NO_2$ pollution, and the weather
situation is high-pressure out-of-sea type and high-pressure control type. The L3 period is December 11,




2021 Beijing time, during which NO$_2$ pollution is carried out and the weather situation is high-pressure controlled. The L4 period is December 15, 2021 Beijing time, during which NO$_2$ pollution and the weather situation is high-pressure out of the sea (Table 2). The simulation performance of uEMEP is reflected by simulating the pollution of NO$_2$ under different weather patterns, and the simulation

performance evaluation formula is detailed in Appendix A.

**Table 2: Case classification.**

| Case number | Case type | Weather type | Date | Primary contaminant | Concentration (μg m$^{-3}$) |
|---|---|---|---|---|---|
| | | | 2021/1/13 | NO$_2$ | 88 |
| L1 | Compound pollution | High pressure control + front impact | 2021/1/14 | PM$_{2.5}$ | 101 |
| | | | 2021/1/15 | NO$_2$ | 137 |
| | | | 2021/1/16 | PM$_{2.5}$ | 116 |
| L2 | Pollution incidents | High-pressure going to sea + high-pressure control | 2021/1/19 | NO$_2$ | 87 |
| | | | 2021/1/20 | | 125 |
| L3 | Single contamination | High voltage control | 2021/12/11 | NO$_2$ | 82 |
| L4 | | High-pressure access to the sea | 2021/12/15 | NO$_2$ | 88 |

## 3 Results

### 3.1 Evaluation of the meteorological simulation performance

In order to verify the simulation ability and reliability of the model, the observed data and the simulated data were used to compare and verify, which are detailed in Appendix A. As can be seen from Table B1 of Appendix B, the 2m temperature simulation for the L2 case is slightly overestimated (0.6 K>±0.5 K), but the *IOA* is compliant(*IOA*≥0.8), and the 2m temperature simulation for L1, L3, and L4 simulations

are all compliant. The L1 simulated relative humidity has a deviation of -1.6 g kg$^{-1}$, but the *IOA* meets the standard(*IOA*≥0.6). There is a certain deviation (0.5, 0.4, 0.1) in the *IOA* of L2, L3 and L4 simulations, but the comparison of the relative humidity of the three pollution periods shows that there is a significant correlation (0.7-0.8) between the hourly observed values and the simulated values, which indicates the rationality of the simulation. In terms of wind speed, although the *R* is not high and the *MB*

is somewhat overestimated (>±0.5 m s$^{-1}$), both *RMSE* and *IOA* are within the standard range(*RMSE*≤2.0 m/s, *IOA*≥0.6). In addition, the temporal series changes of meteorological conditions are analyzed, and the model performance can reproduce the temporal and spatial changes of meteorological conditions well (Fig. 4). Therefore, the WRF model is reliable for meteorological results for the four pollution periods.



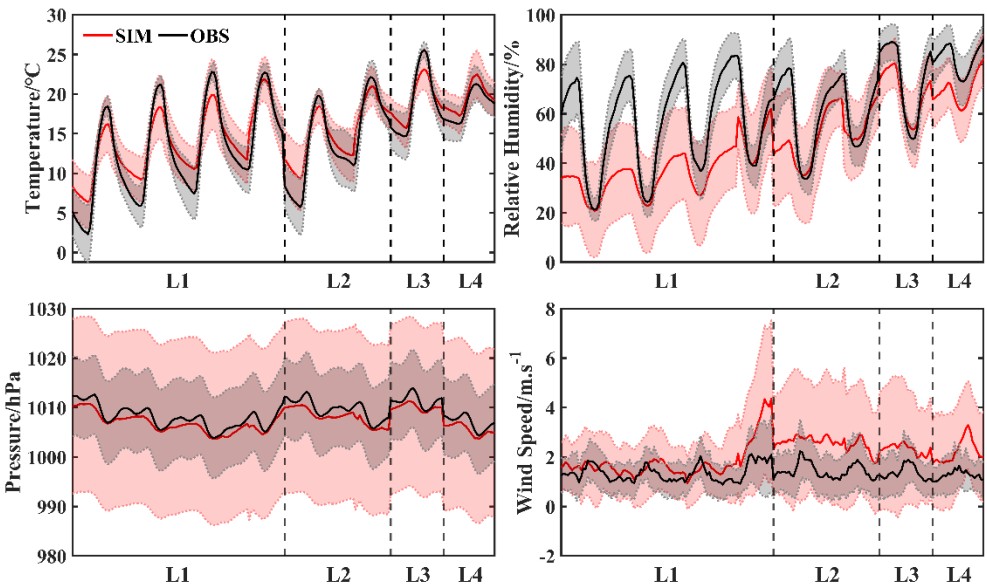


**Figure 4: Shadow plots of simulation and observed error in 4 cases, where the shaded part represents the standard deviation.**

## 3.2 Evaluation of the air quality simulation performance

As can be seen from Table B2 ~ Table B5 of Appendix B, the NME of $NO_2$ and $PM_{2.5}$ in the four cases simulated by EMEP reached the standard($NO_2$:*NME*<80%, $PM_{2.5}$:*NME*<150%), the NME of $NO_2$ and
$PM_{2.5}$ in the L1, L2 and L3 cases simulated by uEMEP reached the standard($NO_2$:*NME*<80%, $PM_{2.5}$:*NME*<150%), the R of $NO_2$ and $PM_{2.5}$ simulated by EMEP reached the standard (0.3~0.6, 0.4~0.5), and the R of L2 and L4 simulated by uEMEP reached the standard (0.3~0.6, 0.3~0.4), the R of $O_3$ simulated by EMEP and uEMEP reached the standard(*R*>0.4), and there was a strong correlation (0.6~0.9). Comparing the simulation performance of EMEP and uEMEP models, it is found that for
$NO_2$ and L4, uEMEP simulation performance is better, but for other cases, the standard deviation and correlation coefficient of uEMEP simulation $O_3$ in four cases are better than those of EMEP model, for $PM_{2.5}$, the simulation performance of uEMEP in L4 is better, but the simulation performance of uEMEP in other cases is poor, for $PM_{10}$, L1, L2, The EMEP simulation performance is better in L3 and the uEMEP simulation performance is better in L1 (Fig. 5).
In general, the performance of the uEMEP model in the L1 case is weaker than that of the EMEP model, the simulation of uEMEP for $NO_2$, $O_3$ and $PM_{10}$ in the L2 case is better than EMEP, the simulation of uEMEP for $O_3$, $PM_{2.5}$ and $PM_{10}$ in the L3 case is better than that of EMEP, and the simulation of uEMEP for $O_3$ and $PM_{10}$ in the L4 case is better than that of EMEP. The overall simulation deviation of uEMEP is large, $NO_2$ is between -95%~-58%, $O_3$ is between -57%~33%, $PM_{2.5}$
is between -92%~-70%, and $PM_{10}$ is between -93%~-76%.



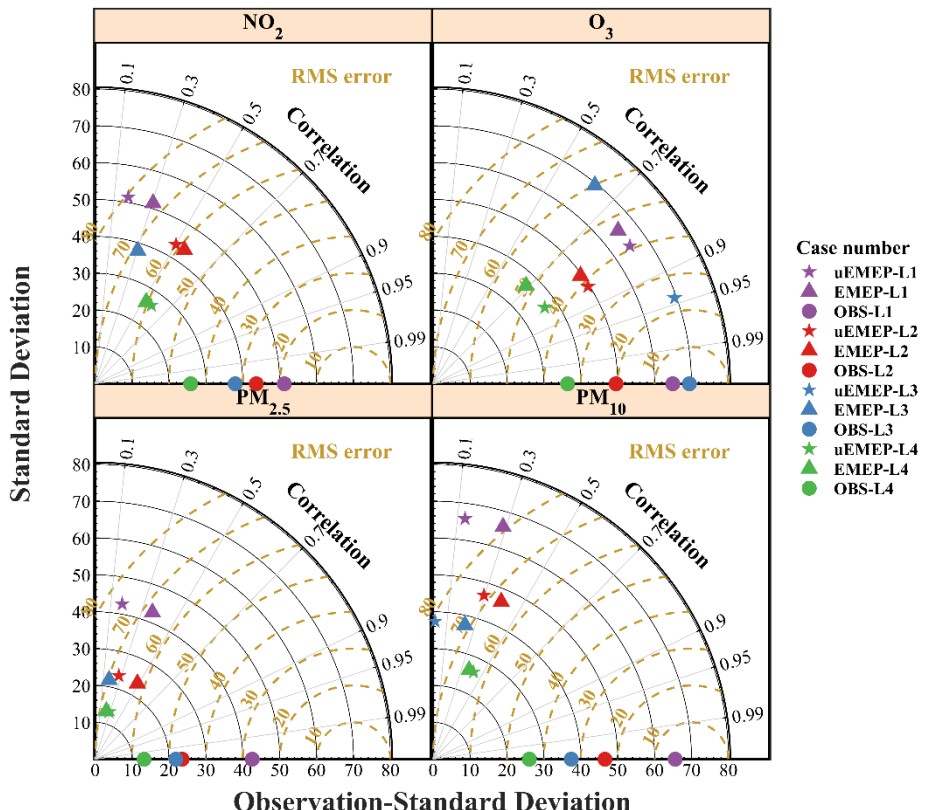

**Figure 5: Comparison of NO₂, O₃, PM₂.₅ and PM₁₀ between EMEP and uEMEP models.**

## 3.3 Spatial characteristics of air pollution

In case L1,From January 13 to January 16, 2021, a compound pollution event occurred in Foshan
City, with the high-pressure control type as the dominant weather type and the frontal influence type as
the secondary weather type. The daily average concentrations of $NO_2$ and $PM_{2.5}$ reached 88-137 µg m⁻³
and 101-116 µg m⁻³, respectively (Table 2).

From January 13th to January 15th, the high-altitude and low-altitude circulation configuration under
the control of high pressure caused the vertical stratification of the atmosphere to tend to be stable, and
the low temperature and low humidity conditions are not conducive to the photochemical conversion of
$NO_2$. On January 16, the weather conditions of the front caused the air in front of the cold front to
accumulate air, and the convergence of the wind field hindered the upward diffusion and outward
transport of local pollutants in Foshan. The spatial distribution of $NO_2$ follows the road network setting.
The high value of $NO_2$ occurred at 20:00 on January 14 (Fig. 6(a)), and the high value area of $NO_2$ was
S263 and S81 in Nanhai District, Guangzhou-Foshan junction in the northeast of Foshan City(Fig. 6(b)-
Fig. 6(c)). From the evening of January 14 (19:00-23:00) to the early morning of January 15 (00:00-
08:00), a large amount of $NO_2$ was accumulated, and the traffic pressure in Foshan during the morning
and evening rush hours was high, resulting in $NO_2$ pollution on January 15.



NO$_2$ is the primary pollutant in the early stage of the pollution process, and the enhancement of
atmospheric oxidation is one of the reasons for the increase of PM$_{2.5}$ concentration. The spatial
distribution of PM$_{2.5}$ pollution is mainly affected by industrial point sources, followed by road network
settings. The high PM$_{2.5}$ value occurred at 12:00 on January 16(Fig. 6(d)), and the wind speed was small
during the day, which was not conducive to the diffusion of high-concentration PM$_{2.5}$ industrial point
sources, resulting in local accumulation of PM$_{2.5}$(Fig. 6(e)- Fig. 6(f)). On January 14, the convergence
of wind fields during the day was not conducive to the spread of PM$_{2.5}$, and PM$_{2.5}$ continued to
accumulate, resulting in PM$_{2.5}$ pollution.

The composite pollution event was mainly affected by meteorological conditions, and regional
transmission accounted for a large proportion. At the same time, Foshan's local industries gradually
began to resume production and work in January(Foshan Emergency Management Bureau and Office,
2021), resulting in an increase in local emissions, resulting in a combined NO$_2$ and PM$_{2.5}$ pollution
incident.





**(a)**

**NO₂ simulation results of L1 pollution period at 5 typical stations**

**(b)** **(c)**

**(d)**

**PM₂.₅ simulation results of L1 pollution period at 5 typical stations**

**(e)** **(f)**




**Figure 6: The time series simulation results of NO₂ and PM₂.₅, and spatial distribution of uEMEP simulation at the peak moment (red hollow circle) in L1 case. (a): Time series of NO₂ simulation results (shaded part is the standard deviation of 5 typical stations). (b) Spatial distribution of NO₂ simulated by EMEP at 20:00 on January 14, 2021. (c) Spatial distribution of NO₂ simulated by uEMEP at 20:00 on January 14, 2021. (d) Time series of PM₂.₅ simulation results(shaded part is the standard deviation of 5 typical stations). (e) Spatial distribution of PM₂.₅ simulated by EMEP at 20:00 on January 14, 2021. (f) Spatial distribution of PM₂.₅ simulated by uEMEP at 20:00 on January 14, 2021.**


In case L2, continuous NO$_2$ pollution occurred on January 19, 2021 and January 20, 2021 in Foshan City under the combined action of high-pressure controlled weather and high-pressure sea-moving weather. The average daily concentration of NO$_2$ reached 87-125 μg m$^{-3}$ (Table 2). The spatial distribution of NO$_2$ follows the road network setting. The high value of NO$_2$ occurred at 20:00 on January 20(Fig. 7(a)), and the high value of NO$_2$ was in the Sanshui District and Nanhai District in the north of Foshan City, and the traffic pressure was greater (Fig. 7(b)- Fig. 7(c)). The ventilation coefficient was small under the high-pressure weather conditions on January 19, and the near-surface pressure gradient decreased and the wind speed weakened under the high-pressure controlled weather conditions on January 20, which was not conducive to the horizontal transportation of air pollutants and hindered the upward and outward transmission of local pollutants in Foshan City.

This compound pollution event is mainly affected by meteorological conditions, and the turbulence is weak, which is not conducive to the horizontal transport of air pollutants and hinders the upward and outward transmission. At the same time, Foshan's local industries gradually began to resume production and work in January, resulting in an increase in local emissions, resulting in a continuous NO$_2$ pollution incident.



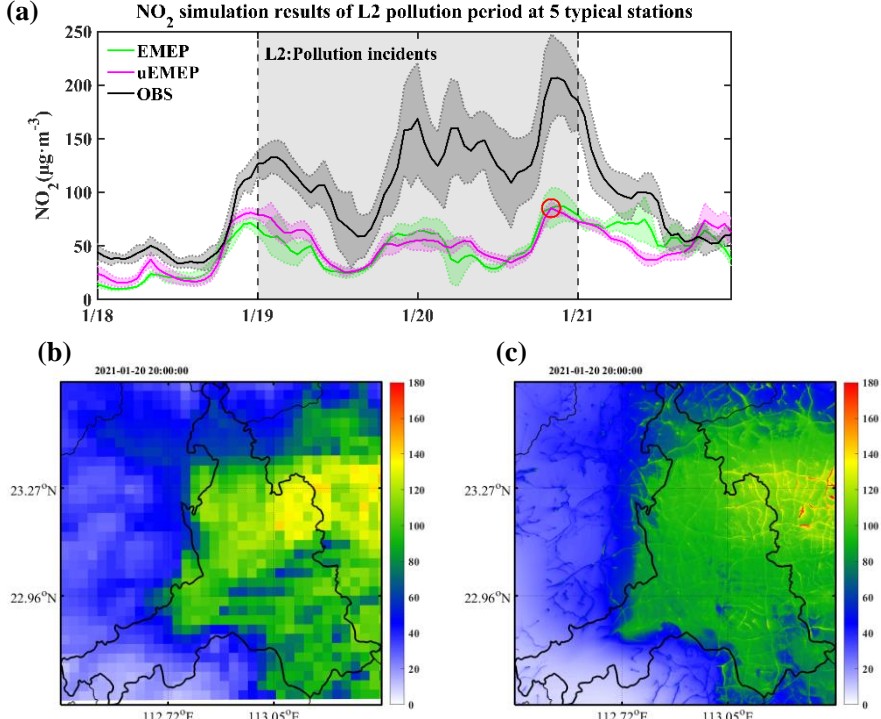

**Figure 7: The time series simulation results of NO₂, and spatial distribution of uEMEP simulation at the peak moment (red hollow circle) in L2 case. (a): Time series of NO₂ simulation results (shaded part is the standard deviation of 5 typical stations). (b) Spatial distribution of NO₂ simulated by EMEP at 20:00 on January 20, 2021. (c) Spatial distribution of NO₂ simulated by uEMEP at 20:00 on January 20, 2021.**

In case L3, on December 11, 2021, a NO₂ pollution occurred in Foshan City with high pressure control as the dominant weather. The average daily concentration of NO₂ reached 82 μg m⁻³ (Table 2). The high value of NO₂ occurred at 19:00 on December 11(Fig. 8(a)), and the high value of NO₂ was concentrated in the S81 and S55 areas at the junction of Guangzhou-Foshan in the central and eastern part of Foshan City (Fig. 8(b)- Fig. 8(c)), and the traffic pressure was greater. Under the high-pressure controlled weather conditions on December 11, the pressure gradient near the surface decreased, and the wind speed weakened, which was not conducive to the migration and dispersion of pollutants. At the same time, the density of local vehicles in Foshan City increased during the morning and evening rush hours, which intensified the traffic pressure and caused the pollution incident.

In case L4, On December 15, 2021, a NO₂ pollution occurred in Foshan City, where high pressure was the dominant weather. The average daily concentration of NO₂ reached 88 μg m⁻³ (Table 2). The high value of NO₂ occurred at 00:00 on December 15(Fig. 8(a)), and the high value of NO₂ was concentrated in the S81 and S55 areas at the junction of Guangzhou-Foshan in the central and eastern part of Foshan City (Fig. 8(d)- Fig. 8(e)). On December 15, the ventilation coefficient was small and the wind speed weakened, which hindered the upward diffusion and outward transport of local pollutants in Foshan, and the NO₂ concentration continued to accumulate on the night of the 14th (20:00-23:00), triggering a pollution incident.



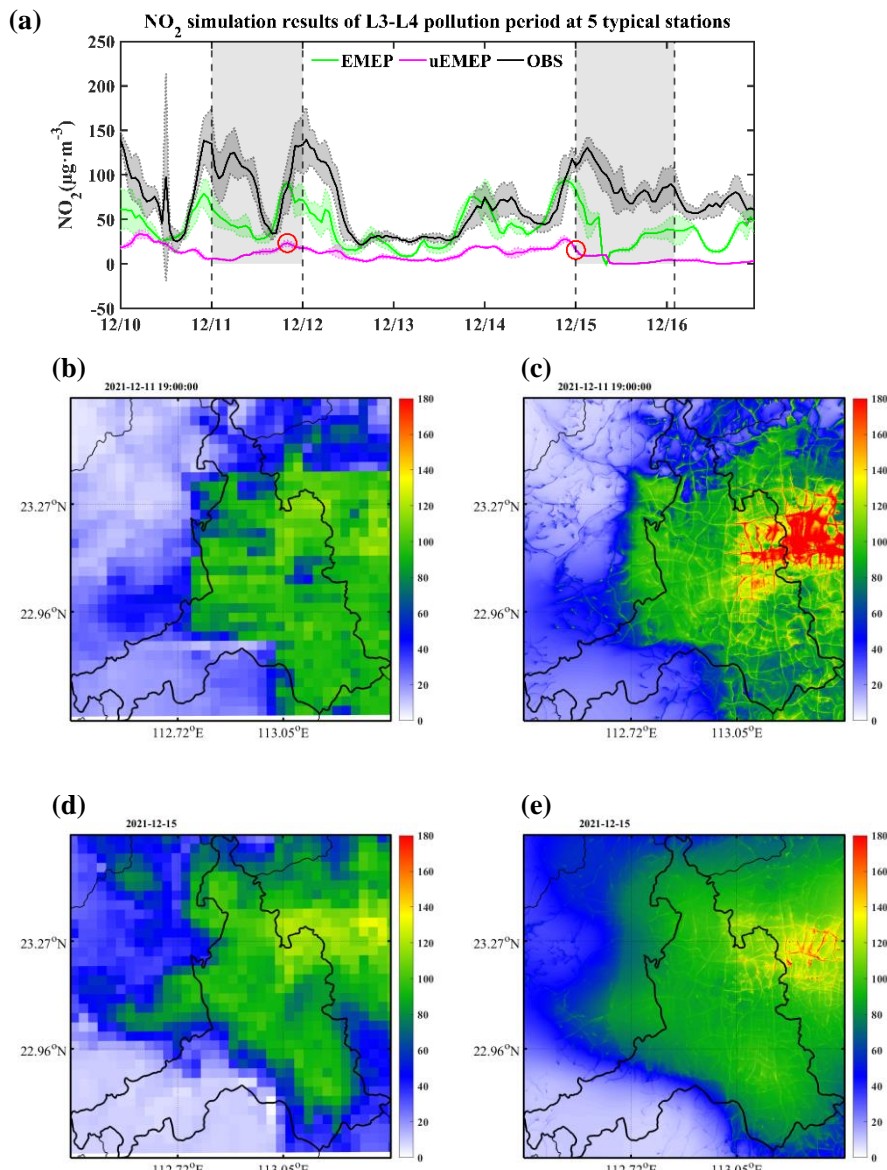


**Figure 8: The time series simulation results of NO₂, and spatial distribution of uEMEP simulation at the peak moment (red hollow circle) in L3 and L4 case. (a): Time series of NO₂ simulation results (shaded part is the standard deviation of 5 typical stations). (b) Spatial distribution of NO₂ simulated by EMEP at 20:00 on December 11, 2021. (c) Spatial distribution of NO₂ simulated by uEMEP at 20:00 on December 11, 2021. (d) Spatial distribution of NO₂ simulated by EMEP at 00:00 on December 15, 2021. (e) Spatial distribution of NO₂ simulated by uEMEP at 00:00 on December 15, 2021.**


Combined with the analysis of the weather circulation situation, compared with the high-pressure weather pattern of L2 and L4 with relatively quiet and small winds, we find that L1 and L3 are sudden and transient weather patterns, especially in the late stage of L1, there is a cold front transit, and the



meteorological elements suddenly change, and the PM$_{2.5}$ concentration near the surface rises through
regional transmission, but the WRF meteorological model is difficult to capture the sudden turbulence
phenomenon, which makes the meteorological field input to the EMEP model have great uncertainty,
and at the same time, it is difficult for uEMEP to restore the pollutant diffusion phenomenon caused by
sudden weather phenomena based on the Gaussian plume principle, so the uEMEP simulation
performance of L1 and L3 is poor.
After the fine simulation deployment with high spatiotemporal resolution in Foshan, comparing the
spatial distribution of NO$_2$ between the EMEP model and the uEMEP model, it can be seen that the
application of the block scale in Foshan can more accurately simulate the distribution and change of air
pollutant concentration, and put forward more scientific and targeted action plans to alleviate air
pollution.

**3.4 Analysis of NO$_2$ traceability characteristics**

In order to quantitatively analyze the sources of NO$_2$ pollution in four cases in different local areas of
Foshan City, and identify the sources of pollutants affecting the ambient air quality monitoring stations
in each city, Sanshuiyundonghai (SSYDH, suburban station), Jinjiju (JJJ, industrial station),
Huacaizhizhong (HCZZ, urban-developed station), Huijingcheng (HJC, urban-developing station), and
Wenhualubianzhan (WHLBZ, roadside station) were selected for analysis(Ecological and
Environmental Monitoring Centre of Guangdon et al., 2023; Liu et al., 2011).
The two urban stations (HJC, HCZZ) are located in the center of Chancheng District, Foshan City, and
are affected by multiple pollution sources such as traffic, industry and residents. Comparing the two
urban sites, we find that the average proportions of HJC regional transmission, shipping emissions and
industrial emissions in these four cases are 87.3%, 1.6% and 0.6%, respectively, which is slightly larger
than that of HCZZ (85%, 1.4% and 0.2%), and the average proportion of HJC industrial emissions in
case L1 reaches 1%, which is greater than that of HCZZ (0.3%). JJJ is located in Shunde District, the
concentration level of high-tech enterprises is higher than the overall level of Foshan City and continues
to grow(Bo et al., 2023), its average shipping emissions account for 2.6%, industrial emissions account
for 0.8%, and residential emissions account for 3%, and the average proportion of JJJ industrial
emissions in case L1 is 1.1%, and the proportion of residential emissions is 3.3%, which is larger than
the other four stations. SSYDH is located in Sanshui District, which is a representative station in the
suburbs, and its average proportion of shipping emissions is 3.6%, which is larger than that of the other
four stations, of which the average proportion of shipping emissions in case L1 reaches 5%, and the
average proportion of industrial emissions and residential emissions is smaller than that of the other four
stations. WELBZ is located in Chancheng District, which has a complex transportation network, and the
average proportion of traffic emissions is 10.1%, which is larger than that of HJC (9.4%). Case L3 and
Case L4 found that the average proportion of regional transmission between the five sites was between
96% and 99.4%, and regional transmission was the primary reason for the occurrence of these two
cases.









**Figure 9: The traceability of NO₂ in the four cases of 5 representative air quality monitoring stations.(a): The traceability of NO₂ in the four cases of HJC (urban-developing station).(b): The traceability of NO₂ in the four cases of HCZZ ( urban-developed station).(c): The traceability of NO₂ in the four cases of JJJ(industrial station).(d): The traceability of NO₂ in the four cases of SSYDH(suburban station).(e): The traceability of NO₂ in the four cases of WHLBZ(roadside station).**


## 4 Discussions and recommendation

### 4.1 Parameterization mechanism need improvement

The urban canopy data used in the WRF model has always had a single representation and ambiguous parameters for the Pearl River Delta region, and the urban canopy is an important factor affecting the
wind field simulation and surface energy balance of the urban underlying surface(Chen et al., 2011). Foshan is an important part of the Pearl River Delta urban agglomeration with a high degree of urbanization, and it is expected that the use of refined urban canopy parameters(Chen et al., 2021) to localize the improvement of Foshan's urban structure in the WRF model and update the height and proportion of urban buildings will definitely optimize the meteorological simulation of Foshan by the
WRF model.
The complex urban structure (building form, roads, green space, etc.) of Foshan has changed the physical parameters of the underlying surface (such as albedo, specific heat capacity, thermal emissivity, etc.), and the rapid urbanization process of Foshan has had a complex and drastic impact on the land-atmosphere exchange process, thus changing the diffusion conditions and chemical reaction
conditions of air pollutants in the city. The chemical mechanism in the EMEP model is EmChem09, which was developed and perfected between 2008 and 2009(Simpson, 2014), but the photochemical reaction module contained in it is simple and difficult to reflect the complex chemical reaction conditions in Foshan. This makes the EMEP model have certain limitations for the simulation of secondary pollutants.

### 4.2 Local emission source and reallocation mechanism need update

The use of China's 2017 monthly average multi-resolution emissions inventory (MEIC) data (27km×27km) to replace the European emission source data in EMEP still has some imprecision in the spatial distribution, although the calculation method mapped to the high-resolution grid is used to ensure that the local emission characteristics remain unchanged after the inventory is replaced, for
example, the list of industrial sectors in the inventory does not include enterprises below designated size (industrial enterprises with main business income of less than 20 million yuan) , causing the simulation to fall below the observed threshold. At the same time, the road traffic emissions obtained from OSM do not take into account city-specific traffic congestion or other possible spatial characteristics, and the emissions of enterprises below designated size are also not included in the industrial point source data
of Foshan. Therefore, the optimization of possible emission inventories can improve the simulation performance of the EMEP model in high-industrial density megacities to a certain extent, so as to further optimize the simulation performance of the uEMEP model.





## 5 Conclusion

In the past, the air quality forecast of Foshan mainly focused on the prediction and simulation of various
air pollutant concentrations at the mesoscale, without considering the impact of urban structure diversity
on the dispersion of local pollutants. In this study, a high-precision air quality model system was built in
Foshan based on the uEMEP model method, and the application of the block-scale air quality model in
Foshan City, a megacity with high-density industrial development, was realized.
The study used meteorological and air quality data to evaluate model performance, classify four
pollution cases, and analyze their spatial characteristics and nitrogen dioxide traceability. The results
show that the uEMEP model can well simulate the distribution of air pollutants at the block scale,
providing insights for targeted air pollution control measures.
Air Quality Simulation Performance The European Monitoring and Evaluation Programme (EMEP) and
its City Scale Counterpart Programme (uEMEP) have demonstrated varying degrees of success in
different pollutants and cases. It is worth noting that the uEMEP model performs well in simulating
$NO_2$ and $O_3$ in a variety of scenarios, surpassing EMEP in terms of spatial resolution and correlation
with observed data, but the overall deviation is large and needs further improvement.
Every pollution event is affected by a combination of meteorological factors and pollution emissions.
The spatial distribution of $NO_2$ closely follows the distribution of urban road networks, and industrial
emissions significantly affect the spatial pattern of pollutants, highlighting the intertwined relationship
between human activities and air quality.
Through a detailed analysis of the traceability characteristics of $NO_2$ in Foshan, the impacts of regional
transportation, transportation, and industrial emissions on different monitoring stations were revealed. It
is clear that urban center suffer from a complex emission profile, while suburbs may be more vulnerable
to shipping activities.
Refining the urban canopy parameters of the WRF model to reflect Foshan's unique urbanization
process can enhance the meteorological simulation. Incorporating emissions from small businesses and
taking into account city-specific traffic attributes can improve the simulation accuracy of EMEP and
uEMEP models. Optimizing emissions inventories is essential to better reflect air quality in
industrialized, densely populated urban areas. In conclusion, multi-scale air quality models are an
indispensable tool for us to interpret the complex dynamics of urban air pollution. Through careful
evaluation and continuous improvement, they lead us towards cleaner and healthier urban
environments.



**Code and data availability**

The uEMEP_v6 model used in this study is archived on Zenodo
https://doi.org/10.5281/zenodo.10708389(Yang Liting, 2024a).The latest development of uEMEP_v5
model can be downloaded at https://github.com/metno/uEMEP (last access: 8 January 2024, Norwegian
Meteorological Institute). The EMEP model version rv4.33 used in this study can be found at
https://github.com/metno/emep-ctm (last access: 8 January 2024, Norwegian Meteorological Institute).
The configuration file of the EMEP model is archived at
https://doi.org/10.5281/zenodo.10703324(Yang Liting, 2024b), as are MATLAB scripts for
visualisation is archived at https://doi.org/10.5281/zenodo.10714310(Yang Liting, 2024c).The codes
and datasets in this publication are available to the community, and they can be accessed by request to
the corresponding author.

**Author contributions:** LY (Writing – original draft preparation), MC (Conceptualization,
Methodology), SS (Data curation, Resources), WW (Writing – review \& editing), XW (Supervision,
Project administration)

**Competing interests:** The authors declare that they have no conflict of interest.

**Acknowledgements:** This work was supported by the National Natural Science Foundation (42275107,
42230701, 42121004, 41705123), the national Key Research and Development Program of China
(2023YFC3706202), the Science and Technology Projects in Guangzhou (2023A04J0251), the Special
Fund Project for Science and Technology Innovation Strategy of Guangdong Province (Grant
No.2019B121205004). The authors thank the High-performance computing platform of Jinan
University, the Research Center of Low Carbon Economy for Guangzhou Region of Jinan University
for their guidance. Road network map data are copyrighted by OpenStreetMap contributors and
available from https://www.openstreetmap.org (last access: 23 February 2024).



## Appendix A

In order to verify the simulation ability and reliability of the model, the observed data and the simulated data were used to compare and verify. The deviation (*MB*), mean absolute error (*MAE*), root mean square error (*RMSE*), standardized mean deviation (*NMB*) and standardized mean error (*NME*) were used to reflect the deviation between the simulated value and the observed value, and the correlation coefficient (*R*) and fitting index (*IOA*) were selected to evaluate the relationship between the simulated

value and the observed value. *MB*, *MAE*, *RMSE*, *NMB* and *NME* are dimensionally statistical, while *R* and *IOA* are dimensionless statistics. In this study, the following criteria were used to judge the performance of meteorological condition simulation(Emery and Tai, 2001; Tesche et al., 2002), with *MB*≤±0.5K and IOA≥0.8 for 2m temperature, *MB*≤±1 g kg$^{-1}$ and *IOA*≥0.6 for humidity, and *MB*≤±0.5 m s$^{-1}$, *RM*SE≤2.0 m s$^{-1}$ and *IOA*≥0.6 for 10m wind speed. The evaluation criteria for NO$_2$ simulation were

-40%<*NMB*<50%, *NME*<80%, *R*>0.3; for O3 simulation, -15%<*NMB*<15%, *NME*<35%, *R*>0.4 ; for PM2.5 simulation, -50%<*NMB*<80%, *NME*<150%, *R*>0.3.

The specific formula for calculating the above statistics is as follows Eq. (3)-Eq. (9):

$$MB = \frac{1}{N}\sum_1^N(SIM_i - OBS_i) \tag{3}$$

$$MAE = \frac{1}{N}\sum_1^N|SIM_i - OBS_i| \tag{4}$$

$$RMSE = \sqrt{\frac{1}{N}\sum_1^N(SIM_i - OBS_i)^2} \tag{5}$$

$$NMB = \frac{\sum_1^N(SIM_i-OBS_i)}{\sum_1^N(OBS_i)} \times 100 \tag{6}$$

$$NME = \frac{\sum_1^N|SIM_i-OBS_i|}{\sum_1^N(OBS_i)} \times 100 \tag{7}$$

$$R = \frac{\sum_1^N(SIM_i-\overline{SIM})(OBS_i-\overline{OBS})}{\sqrt{\sum_1^N(SIM_i-\overline{SIM})^2}\sqrt{\sum_1^N(OBS_i-\overline{OBS})^2}} \tag{8}$$

$$IOA = 1 - \left[\frac{n \cdot RMSE^2}{\sum_{i=1}^n(|SIM_i|+|OBS_i|)^2}\right] \tag{9}$$

where i=1,2,...N; N is the length of the sequence; SIM$_i$ and OBS$_i$ are the average values of the simulation and observation data; and is the average value of the simulation and observation data, which is calculated by the formula $\overline{SIM} = \frac{\sum_1^N SIM_i}{N}$ and $\overline{OBS} = \frac{\sum_1^N OBS_i}{N}$ .




## Appendix B

**Table B1: Verification of meteorological factors of WRF model in 4 cases.**

| Case number | Meteorological factors | OBS | SIM | MB | MAE | RMSE | R | IOA |
|---|---|---|---|---|---|---|---|---|
| L1 | Temperature(°C) | 13.9 | 14.3 | 0.4 | 3.2 | 3.8 | 0.8 | 0.9 |
| | Relative Humidity(%) | 55.9 | 38.7 | -17.2 | 23.6 | 28.0 | 0.5 | 0.6 |
| | Pressure(hPa) | 1008.1 | 1007.0 | -1.1 | 8.8 | 8.8 | 0.9 | 0.9 |
| | Wind Speed(m s⁻¹) | 1.5 | 2.0 | 0.6 | 1.4 | 1.8 | 0.4 | 1.0 |
| L2 | Temperature(°C) | 15.7 | 16.3 | 0.6 | 2.2 | 2.7 | 0.8 | 0.9 |
| | Relative Humidity(%) | 61.4 | 55.7 | -5.7 | 14.7 | 18.1 | 0.7 | 0.5 |
| | Pressure(hPa) | 1008.9 | 1007.4 | -1.6 | 8.6 | 8.7 | 0.9 | 0.8 |
| | Wind Speed(m s⁻¹) | 1.4 | 2.4 | 1.0 | 1.7 | 1.9 | 0.2 | 0.9 |
| L3 | Temperature(°C) | 19.2 | 19.2 | 0.0 | 2.3 | 2.7 | 0.7 | 0.9 |
| | Relative Humidity(%) | 74.4 | 68.2 | -6.2 | 12.2 | 14.6 | 0.8 | 0.4 |
| | Pressure(hPa) | 1011.5 | 1010.4 | -1.1 | 8.6 | 8.7 | 0.7 | 0.6 |
| | Wind Speed(m s⁻¹) | 1.3 | 2.4 | 1.1 | 1.7 | 2.0 | 0.1 | 0.9 |
| L4 | Temperature(°C) | 19.2 | 19.8 | 0.5 | 2.5 | 2.8 | 0.7 | 0.8 |
| | Relative Humidity(%) | 84.8 | 75.5 | -9.3 | 13.1 | 15.0 | 0.8 | 0.1 |
| | Pressure(hPa) | 1006.5 | 1005.1 | -1.4 | 8.5 | 8.6 | 0.8 | 0.6 |
| | Wind Speed(m s⁻¹) | 1.3 | 2.3 | 1.0 | 1.6 | 1.8 | 0.1 | 0.9 |


**Table B2: The EMEP and uEMEP models simulate NO₂ performance.**

| Case number | NO₂(μg/m³) | EMEP | uEMEP |
|---|---|---|---|
| L1 | OBS | 141.3 | |
| | SIM | 54.1 | 54.6 |
| | MB | -88.0 | -87.1 |
| | NMB | -60.5 | -59.3 |
| | NME | 61.3 | 60.9 |
| | R | 0.3 | 0.2 |
| L2 | OBS | 119.4 | |
| | SIM | 49.8 | 51.2 |
| | MB | -70.2 | -68.7 |
| | NMB | -57.9 | -56.8 |
| | NME | 58.7 | 58.2 |
| | R | 0.6 | 0.5 |
| L3 | OBS | 96.7 | |
| | SIM | 49.5 | 13.3 |
| | MB | -47.6 | -83.4 |
| | NMB | -46.9 | -85.0 |
| | NME | 53.7 | 85.3 |





| | | | |
|---|---|---|---|
| | **R** | 0.3 | -0.2 |
| **L4** | **OBS** | 92.5 | |
| | **SIM** | 34.0 | 5.0 |
| | **MB** | -58.9 | -87.5 |
| | **NMB** | -59.8 | -90.5 |
| | **NME** | 60.7 | 90.5 |
| | **R** | 0.5 | 0.6 |

**Table B3: The EMEP and uEMEP models simulate $O_3$ performance.**

| Case number | $O_3(\mu g/m^3)$ | EMEP | uEMEP |
|---|---|---|---|
| **L1** | **OBS** | 48.6 | |
| | **SIM** | 27.1 | 21.3 |
| | **MB** | -21.8 | -27.5 |
| | **NMB** | -41.1 | -53.8 |
| | **NME** | 69.2 | 65.8 |
| | **R** | 0.8 | 0.8 |
| **L2** | **OBS** | 40.7 | |
| | **SIM** | 25.7 | 21.2 |
| | **MB** | -14.7 | -19.1 |
| | **NMB** | -32.6 | -44.9 |
| | **NME** | 63.9 | 61.8 |
| | **R** | 0.8 | 0.8 |
| **L3** | **OBS** | 64.4 | |
| | **SIM** | 24.9 | 50.4 |
| | **MB** | -39.3 | -13.9 |
| | **NMB** | -60.6 | -19.5 |
| | **NME** | 74.9 | 75.7 |
| | **R** | 0.6 | 0.9 |
| **L4** | **OBS** | 39.5 | |
| | **SIM** | 33.7 | 53.8 |
| | **MB** | -6.9 | 13.1 |
| | **NMB** | -13.7 | 38.0 |
| | **NME** | 56.1 | 70.3 |
| | **R** | 0.7 | 0.8 |


**Table B4: The EMEP and uEMEP models simulate $PM_{2.5}$ performance.**

| Case number | $PM_{2.5}(\mu g/m^3)$ | EMEP | uEMEP |
|---|---|---|---|
| **L1** | **OBS** | 106.9 | |
| | **SIM** | 24.3 | 27.0 |
| | **MB** | -83.0 | -80.0 |
| | **NMB** | -76.4 | -73.4 |



| | | EMEP | uEMEP |
|---|---|---|---|
| | NME | 76.4 | 73.6 |
| | R | 0.4 | 0.2 |
| **L2** | OBS | 82.9 | |
| | SIM | 20.7 | 25.4 |
| | MB | -62.6 | -57.7 |
| | NMB | -74.9 | -69.0 |
| | NME | 74.9 | 69.0 |
| | R | 0.5 | 0.3 |
| **L3** | OBS | 70.4 | |
| | SIM | 21.0 | 9.9 |
| | MB | -49.6 | -60.6 |
| | NMB | -69.2 | -85.2 |
| | NME | 69.6 | 85.2 |
| | R | 0.2 | 0.0 |
| **L4** | OBS | 71.2 | |
| | SIM | 14.3 | 5.5 |
| | MB | -57.2 | -65.9 |
| | NMB | -76.0 | -88.3 |
| | NME | 76.1 | 88.3 |
| | R | 0.2 | 0.3 |

**Table B5: The EMEP and uEMEP models simulate PM$_{10}$ performance.**

| Case number | PM$_{10}$(μg/m$^3$) | EMEP | uEMEP |
|---|---|---|---|
| **L1** | OBS | 181.5 | |
| | SIM | 51.2 | 41.8 |
| | MB | -131.1 | -140.0 |
| | NMB | -71.0 | -75.8 |
| | NME | 71.2 | 76.0 |
| | R | 0.3 | 0.1 |
| **L2** | OBS | 163.9 | |
| | SIM | 42.6 | 40.0 |
| | MB | -121.9 | -124.3 |
| | NMB | -74.0 | -75.6 |
| | NME | 74.0 | 75.6 |
| | R | 0.4 | 0.3 |
| **L3** | OBS | 131.4 | |
| | SIM | 43.7 | 14.4 |
| | MB | -88.1 | -117.1 |
| | NMB | -66.2 | -88.6 |
| | NME | 66.6 | 88.6 |
| | R | 0.2 | 0.0 |
| **L4** | OBS | 116.8 | |
| | SIM | 31.3 | 8.4 |
| | MB | -86.5 | -109.0 |
| | NMB | -70.4 | -89.3 |
| | NME | 71.9 | 89.8 |
| | R | 0.4 | 0.4 |



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
