# Peer review of "Development And Application of WRF (v4.1.2)-uEMEP(v5) Model at the City with the Highest Industrial Density: A Case Study of Foshan"

_EGUsphere, 2024_

## Author Comment (AC1)

**Respose to the Comments from Reviewer 1**

Thank you for giving me the opportunity to submit a revised draft of my manuscript titled "Development And Application of WRF(v4.1.2)-uEMEP(v5) Model at the City with the Highest Industrial Density: A Case Study of Foshan". We appreciate the time and effort that you and the reviewers have dedicated to providing your valuable feedback on my manuscript. We are grateful to the reviewers for their insightful comments on my paper. We have been able to incorporate changes to reflect most of the suggestions provided by the reviewers. We have highlighted the changes within the manuscript.

Here is a point-by-point response to the reviewers' comments and concerns.

**Comments from Reviewer 1**

Comment1:Abstract: The abstract is repetitive and provides what I would consider unnecessary details for an abstract, but does not specify the main conclusions of this study. I'd recommend only including the most important context in the abstract.

Response: We agree with this comment. Therefore,we modified the abstract "Abstract:The study aims to develop and apply the WRF-uEMEP model to simulate air quality at the urban scale, focusing on Foshan, a city with high industrial density. The model takes into account the impact of urban structure and considers atmospheric dispersion and chemical reactions in different regions. The research process includes model development, calibration, and validation using existing air quality data in Foshan, as well as exploring the characteristics of nitrogen oxide pollution cases under different weather patterns. The study shows that the WRF-uEMEP model effectively captures the impact of urban structure on air pollutant processes. Additionally, the dominant weather patterns for $NO_2$ pollution cases in Foshan are mainly high-pressure control, high-pressure offshore, and frontal influence. Traffic emissions are the primary local source of $NO_2$ pollution in Foshan, accounting for an average of 69.7% of contributions, followed by residential emissions (19.1%), industrial emissions (8.3%), and shipping emissions (2.9%)."

Comment 2:Introduction: The author mentioned many models in this part. It would be helpful to provide their full names for us to understand the application of each model, especially the EMEP that mainly used in this study. Please provide its full name, and what does the "u" in "uEMEP" represent?

Response: Thank you for this suggestion. I have added full names of each model,such as "The Enviro-HIRLAM-M2UE" to "The Enviro-HIRLAM-M2UE(Environment- HIgh Resolution Limited Area Model-Micro scale Model for Urban Environment) model"; "CMAQ-ADMS(Roads)" to "The Community Multiscale Air Quality modelling system and the Atmospheric Dispersion Modelling System (CMAQ-ADMS(Roads))"; "WRF-Chem" to "Weather Research and Forecasting model with Chemistry model"; "KC-TRAQS" to "The Kansas City TRansportation local-scale Air Quality Study"; "LUR" to "the Land Use Regression"; "The GEM-MACH-PAH" to "(Global Environment Multiscale Modelling Air quality and Chemistry- Polycyclic Aromatic Hydrocarbon)"; "HYCAMR" to "The hybrid modeling framework"; "CAMx" to "the Comprehensive Air Quality Model with

Extensions"; "CAIRDIO-Les" to "The LES microscale simulations with the topography-resolving urban dispersion model CAIRDIO (CAIRDIO-Les)"; "uEMEP" to "The urban EMEP (uEMEP)"; "EMEP MSC-W" to "European Monitoring and Evaluation Programme Meteorological Synthesising Centre West".

Comment 3:line 99: Please provide the website address for "data GLC2020 the European Space Agency (ESA)". Additionally, there is a grammar issue with this sentence.

Response: Thank you for this suggestion. I have added the website address" land use data dataset uses Copernicus (ECMWF) GLCs2020 satellite observation 300-meter resolution grid data (https://cds.climate.copernicus.eu/cdsapp#!/dataset/satellite-land-cover)"

Comment 4:Figure 2: It is difficult to see whether the content represented by the second legend "Districts in Foshan" has already been displayed in the figure. Grey outline or black outline?

Response: Thank you for this suggestion. I modified the color of the borders of Foshan (black) and the borders of each administrative region of Foshan (yellow) in the Figure 2.

Comment 5:line 115-123: Please improve the description of these two methods and what are their respective characteristics? What are the similarities and differences between them, and what's the meaning of the sentence "the proxy data is given in the form of emissions and summarized into the CTM grid emissions, and the two methods are equivalent"? How do we understand the meaning of "equivalent"? Why did the author choose the first method and what are its advantages?

Response: Thank you for pointing this out. I modified the statement about the downscaling method.

Original copy: The uEMEP mode can be run using two downscaling methods, both of which utilize a Gaussian diffusion model to simulate the concentration of contaminants at high resolution. The choice of downscaling method will depend on high-resolution emissions data, the first of which is the emissions redistribution method, which means that only the following types of high-resolution emissions data are available, such as population density, road network data, or land-use data. The second downscaling method, the independent emission method, is available in both uEMEP and EMEP modes of input high spatiotemporal resolution emission inventories, and the gridded emissions data are fully consistent with the local emissions data, i.e., the proxy data is given in the form of emissions and summarized into the CTM grid emissions, and the two methods are equivalent (Mu et al., 2022). Based on the emission redistribution method, the EMEP-uEMEP model can be used to simulate the air quality at the scale of 100-meter urban blocks, and effectively simulate the diffusion of ozone precursors and particulate matter at the scale of urban blocks, which has broad application prospects.

Modified version: The uEMEP model can be run using two downscaling methods, both of which utilize Gaussian diffusion principles to simulate high-resolution pollutant concentrations. The choice of downscaling method will depend on the high-resolution emissions data. The first downscaling method is the emission redistribution method,

which refers to using only high-resolution emission data such as population density, road network data, or industrial point source data as uEMEP model input data, and redistributing local emission data in the uEMEP model.

The second downscaling method is the independent emission method, which means that the input high-resolution emission inventory is suitable for both uEMEP and EMEP models. At this time, the mesoscale gridded emission data in the EMEP model is completely consistent with the local emission data in the uEMEP model. That is to say, the emissions input from the EMEP model into the uEMEP model at this time are the local emissions of the uEMEP model (Mu et al., 2022). Based on the emission redistribution method, the EMEP-uEMEP model can be used to simulate the air quality at the scale of 100-meter urban blocks, and effectively simulate the diffusion of ozone precursors and particulate matter at the scale of urban blocks, which has broad application prospects.

Comment 6:line 116: "contaminants" or "pollutants"?

Response: Thank you for pointing this out.pollutants, We have modified "contaminants" to "pollutants".

Comment 7:line 141: What is the spatiotemporal resolution of the MEIC used in this study? If the monthly mean emission was used in this study, and how to allocate emissions to reflect daily or hourly variation during the study period? In addition, MEIC inventory has been updated to the year 2020, and compared to 2017, the emissions might have changed significantly, especially after COVID-19. It is obvious that using the emissions from 2017 is no longer appropriate for the simulated period of 2021 in this study.

Response: Thank you for this suggestion. The MEIC resolution is $0.25°\times0.25°$. We have added instructions about it:"In this study, the $0.25°\times0.25°$ China's monthly average multi-resolution emission inventory in 2017 (http://meicmodel.org.cn,MEIC2017) data was used to replace the European emission source data in EMEP.".MEIC2017 was chosen because 2020 was during the epidemic and the resumption of work and production in 2021 stimulated pollutant emissions. Although 2020 is close to 2021, the 2017 inventory is more consistent with 2021 than 2020. Characteristics of pollutant emissions after industrial resumption in Foshan.

Comment 8:line 143: What's the "SNAP" method? Please provide the full name and relevant references and describe this method in detail.

Response: Thank you for this suggestion. We have added table descriptions:

**Table 2: Table of redistribution coefficients (using $NO_x$ as an example).**

| SNAP type | Emissions sector | MEIC type | Emissions sector | redistribution coefficient |
|---|---|---|---|---|
| SNAP1 | combustion in energy and transformation industries | MEIC1 | agriculture | SNAP1=MEIC2*0.27 |
| SNAP2 | non-industrial combustion plants | | | SNAP2=MEIC3 |
| SNAP3 | combustion in manufacturing industry | MEIC2 | industries | SNAP3=MEIC2*0.45 |
| SNAP4 | production processes | | | SNAP4=MEIC2*0.28 |
| SNAP5 | extraction and distribution of fossil fuels and geothermal energy | MEIC3 | fixed combustion | SNAP5=MEIC5 |

| SNAP6 | solvent and other product use | | | SNAP6=MEIC3*0 |
| SNAP7 | road transport | MEIC4 | Residention | SNAP7=MEIC5*0.65 |
| SNAP8 | other mobile sources and machinery | | | SNAP8=MEIC5*0.35 |
| SNAP9 | waste treatment and disposal | | | SNAP9=MEIC3*0 |
| SNAP10 | agriculture | MEIC5 | transportation | SNAP10=MEIC3*0 |
| SNAP11 | other sources and sinks | | | SNAP11=MEIC3*0 |

Comment 9:line 150: How to obtain the "allocation coefficient"?

Response: Thank you for pointing this out. We obtained the distribution coefficient by surveying relevant literature statistics, and we have added relevant descriptions to the article.

Comment 10:line 157-165: In my opinion, the emissions in downscaling models should be remapped based on total emissions and higher resolution data, such road network, population, or industry. I do not understand what the process of "replace (line 162)" and "reduce (line 165)" the author mentioned during inventory processing. Please review and describe the inventory processing in detail. The current description is not very clear.

Response: Thank you for pointing this out. I modified the statement about the total emissions and higher resolution data: The main types of emission data that need to be prepared for uEMEP are: traffic, residential combustion, shipping and industry. For the preparation of Foshan's local emission data, this study uses OpenStreetMap (OSM) (Openstreetmap Contributors, 2020) road network data and updates localized road weights to obtain traffic exhaust emission data; inputs a 250m grid population dataset(the Global Human Settlement Layer ,http://data.europa.eu/89h/2ff68a52-5b5b-4a22-8f40-c41da8332cfe) to replace residential combustion emission data; uses the Pearl River Delta localized shipping emission data, and shipping emission data covers Pearl River Delta Main rivers and shipping ports in the triangle; Foshan City's industrial source data is used and industrial source emission data includes NOx and particulate matter (PM2.5, PM10) emissions (Fig. 3).

Comment 11:line 179: Is the headline appropriate? Can you consider using the expression of "polluted periods" or others?

Response: Thank you for pointing this out. We agree with this comment. Therefore, we have usd the expression of "polluted periods"

Comment 12:line 180-190: Some descriptions of meteorological conditions in this part are inconsistent with those in Table 2, for example, "high-pressure out-of-sea" and "High-pressure going to Sea" in L2, "High voltage control" in L3, "high-pressure out of the sea" and "High-pressure access to the sea" in L4. These make me feel really confused.

Response: Thank you for pointing this out. I have changed and unified the expression.

Comment 13:1 Section 3.1: What is the number of simples for the model validation in each case? Has the confidence test been passed? What is the reason for the poor performance of simulated wind speed? Is it related to the selection of parameterization schemes in WRF model? Please explain.

Response: Thank you for this suggestion. Since the weather types differ among

the four cases, an overall assessment of the simulation performance for all four cases has not been conducted. However, here is a supplementary explanation regarding the simulation deviations and correlations for temperature, relative humidity, atmospheric pressure, and wind speed compared to the observed values:

For temperature, the simulation deviation is 0.37°C, and the correlation is 0.75.

For relative humidity, the simulation deviation is -9.6%, and the correlation is 0.7.

For atmospheric pressure, the simulation deviation is -1.3 hPa, and the correlation is 0.8.

For wind speed, the simulation deviation is 0.9 m/s, and the correlation is 0.2.

Please note that these values provide a general indication of the simulation performance but may vary depending on specific conditions and locations within the simulation domain.

The overestimation of wind speed may be because this study did not update the latest Foshan urban canopy parameter data set in the WRF model. In addition, the temporal series changes of meteorological conditions are analyzed, and the model performance can reproduce the temporal and spatial changes of meteorological conditions well (Fig. 4). Therefore, the WRF model is reliable for meteorological results for the four pollution periods.

Comment 14:Figure 5: What's the meaning of the "Observation-Standard Deviation"? How to calculate this? And there are no units in the Figure and caption, please check and revise. Additionally, shouldn't the validation of simulation results be compared with observations? There is no relevant description in the caption. If there are other comparison methods, please explain.

Response: Thank you for pointing this out. I added an explanation of Standard Deviation in the title of the Figure 5: Comparison of $NO_2$, $O_3$, $PM_{2.5}$ and $PM_{10}$ between EMEP and uEMEP models(It includes three evaluation indicators: correlation coefficient (R), root mean square error (RMSE) and standard deviation (STD). Please see Appendix A for the specific calculation formula).

Appendix A:

$$STD_{sim} = \sqrt{\frac{1}{N}\Sigma_1^N(SIM_i - \overline{SIM})^2} \tag{10}$$

$$STD_{obs} = \sqrt{\frac{1}{N}\Sigma_1^N(OBS_i - \overline{OBS})^2} \tag{11}$$

At the same time, I increased the axis units in the Figure 5. The points on the x-axis in the graph are the observed values.

Comment 15:Figures 6,7, 8: There also no units for the special distribution figures.

Response: Thank you for pointing this out. I modified the Figures 6,7, 8.

Comment 16:Section 3.4: What methods are used for the "Analysis of NO2 traceability characteristics"? By using the model results?

Response: Thank you for this suggestion. The uEMEP model can calculate the concentrations of different emissions to derive the contribution of different emission sectors in local emissions.

---

## Author Comment (AC2)

**Respose to the Comments from Reviewer 2**

We would like to thank you for your careful reading, helpful comments, and constructive suggestions, which has significantly improved the presentation of our manuscript. We are grateful to the reviewers for their insightful comments on my paper. We have been able to incorporate changes to reflect most of the suggestions provided by the reviewers. We have highlighted the changes within the manuscript.

Indeed, the uneven distribution of monthly emission factors in the EMEP model could lead to a similar issue in the regional chemical fields downscaled and redistributed to the uEMEP model, resulting in a certain degree of error between the simulation results and observations. Meanwhile, the high-precision industrial emission data used in this study temporarily only considered enterprises above a certain scale in Foshan City, lacking emission data from small and medium-sized industrial enterprises. Incorporating industrial emissions from these smaller enterprises and considering specific urban traffic characteristics could enhance the simulation accuracy of the EMEP and uEMEP models. Optimizing emission inventories is crucial for better reflecting air quality in industrialized and densely populated urban areas. This study explored the localization deployment and simulation performance of the uEMEP model in Foshan City, a region with a high population density, dense industrialization, and a complex road network. While there is still significant room for improvement in the model's performance, its pioneering application in China holds a certain degree of research significance.

Here is a point-by-point response to the reviewers' comments and concerns.

**Comments from Reviewer 2**

Comment 1:The authors say in several places that uEMEP results bring added values compared to EMEP (l. 325…, 400…, 405...). This is not supported by the figures they give in Appendix B. Regarding Normalized Mean Bias for example, uEMEP performs marginally better than EMEP for L1-L2, much worse for L3-L4 (Table B2). The same is true for PM2.5 (Table B4). Therefore, the authors seem to be discussing what the wanted to find (strong added value with uEMEP) rather than what they actually found (no added value / degradation). This appeare to me as a major flaw in the scientific method.

Response: Thank you for pointing this out. Your observation is valid. Since the uEMEP model is driven by the EMEP model, the pollutant spatial fields input into uEMEP from EMEP have a significant impact on its simulation performance. As you mentioned, Figure 4 demonstrates that while both models exhibit relatively large deviations from the observations, the uEMEP model, with its input of higher-resolution emission inventories, does exhibit slightly better simulation performance compared to the EMEP model.This improved performance can be attributed to the fact that the higher-resolution emission inventories provide a more detailed and accurate representation of the pollutant sources and their spatial distribution in the region. This, in turn, allows the uEMEP model to generate more precise simulations of air quality.However, it is important to note that despite the slight improvement, there is still room for further optimization and refinement of both the EMEP and uEMEP models.

As discussed earlier, the inclusion of emissions from smaller industrial enterprises and the consideration of urban-specific factors, such as road networks and population density, could help improve the accuracy of the models even further.

Comment 2:Tables B2 and B4 show a general and massive underestimation by the simulations in both NO2 and PM2.5. This is not discussed in the article, and questions all the results.

Response: Thank you for pointing this out. The high-precision industrial emission data used in this study only considered enterprises above a certain scale in Foshan City, and lacked data on emissions from small and medium-sized industrial enterprises. This could be one of the reasons for the underestimation of $NO_2$ and $PM_{2.5}$ simulations.

Comment 3:The tracing methodology in Figure 9 is not explained, and I find the results very questionable. The authors mention that « regional transmission » (meaning not clearly defined) represents up to 99.4 % of the total NO2 quantity for two pollution peaks. How could this possibly happen in a city like Foshan which is presented by the authors as extremely industrialized and with strong trafic ? Here again, the authors seem to lack a critical analysis of their results.

Response: Thank you for pointing this out. The uEMEP model can calculate the concentrations of different emissions to derive the contribution of different emission sectors in local emissions. Our explanation of the source attribution analysis based on the concept of a "moving window" is clear. The number of sub-grids included in the "moving window" determines the relative weight assigned to external transport and local emissions during the attribution process.

In our case, having set the grid size in the "moving window" to 3x3, it is possible that this configuration resulted in a relatively high proportion of the total pollution attributed to regional transport. However, when focusing on the contribution of local sectors, you observed that the transportation sector had the largest share.

This finding is consistent with the common understanding that in urban areas with dense road networks, the transportation sector is a significant contributor to air pollution. The high local contribution from the transportation sector highlights the importance of targeting this sector for pollution reduction measures in Foshan City.

It is also worth noting that the choice of grid size in the "moving window" can affect the attribution results. Larger grid sizes may capture more regional transport, while smaller grid sizes may provide a more localized perspective. Therefore, sensitivity testing with different grid sizes could provide additional insights into the relative contributions of external transport and local emissions.

Comment 4:What I see in their results, with such a massive underestimation of PM2.5 and NO2 is probably either a massive underestimation in emissions (which the authors seem to consider in their conclusions). The point of Gaussing grid modelling with tools such as uEMEP being to better evaluate benefit of a good knowledge of local emissions to improve street-level results, the fact that the emissions seem to be so massively underestimated questions the entire point of the article.

Response: Thank you for pointing this out. We agree with this comment. Absolutely, the underestimation in the emission inventory is indeed a major contributor to the underestimation in the simulation results. As you mentioned, the lack of emission

data from small and medium-sized industrial enterprises in the current study likely contributed significantly to this underestimation, especially for pollutants like $NO_2$ and $PM_{2.5}$. To improve the accuracy of the simulations, it is crucial to optimize the emission inventory by including emissions from all relevant sources, including smaller industrial enterprises. This will help provide a more comprehensive representation of the emissions landscape in the region and potentially lead to more accurate predictions of air quality.In addition, the complexity of the urban environment, including road networks, population density, and industrialization, also plays a significant role in air quality modeling. Therefore, a comprehensive and updated emission inventory that considers all these factors is essential for achieving accurate simulation results.